# A Fully Inkjet-Printed Strain Sensor Based on Carbon Nanotubes

**Hsuan-Ling Kao [1,2,3,*]**, **Cheng-Lin Cho [1]**, **Li-Chun Chang [4,5]**, **Chun-Bing Chen [6]**, **Wen-Hung Chung [6]** and **Yun-Chen Tsai [1]**

[1] Department of Electronic Engineering, Chang Gung University, Tao-Yuan 33302, Taiwan; iptclc@gmail.com (C.-L.C.); np260369@gmail.com (Y.-C.T.)
[2] Centre for Reliability Sciences and Technologies, Chang Gung University, Tao-Yuan 33302, Taiwan
[3] Department of Dermatology, Chang Gung Memorial Hospital, Linkou Branches, Tao-Yuan 33305, Taiwan
[4] Department of Materials Engineering, Ming Chi University of Technology, New Taipei 243303, Taiwan; lcchang@mail.mcut.edu.tw
[5] Center for Thin Film Technologies and Applications, Ming Chi University of Technology, New Taipei 243303, Taiwan
[6] Department of Dermatology, Chang Gung Memorial Hospital, Keelung, Taipei, and Linkou Branches, Tao-Yuan 33305, Taiwan; chunbing.chen@gmail.com (C.-B.C.); wenhungchung@yahoo.com (W.-H.C.)
\* Correspondence: snoopy@mail.cgu.edu.tw; Tel.: +886-3-211-8800 (ext. 5951)

**Abstract:** A fully inkjet-printed strain sensor based on carbon nanotubes (CNTs) was fabricated in this study for microstrain and microcrack detection. Carbon nanotubes and silver films were used as the sensing layer and conductive layer, respectively. Inkjet-printed CNTs easily undergo agglomeration due to van der Waals forces between CNTs, resulting in uneven films. The uniformity of CNT film affects the electrical and mechanical properties. Multi-pass printing and pattern rotation provided precise quantities of sensing materials, enabling the realization of uniform CNT films and stable resistance. Three strain sensors printed eight-layer CNT film by unidirectional printing, rotated by 180° and 90° were compared. The low density on one side of eight-layer CNT film by unidirectional printing results in more disconnection and poor connectivity with the silver film, thereby, significantly increasing the resistance. For 180° rotation eight-layer strain sensors, lower sensitivity and smaller measured range were found because strain was applied to the uneven CNT film resulting in non-uniform strain distribution. Lower resistance and better strain sensitivity was obtained for eight-layer strain sensor with 90° rotation because of uniform film. Given the uniform surface morphology and saturated sheet resistance of the 20-layer CNT film, the strain performance of the 20-layer CNT strain sensor was also examined. Excluding the permanent destruction of the first strain, 0.76% and 1.05% responses were obtained for the 8- and 20-layer strain sensors under strain between 0% and 3128 με, respectively, which demonstrates the high reproducibility and recoverability of the sensor. The gauge factor (GF) of 20-layer strain sensor was found to be 2.77 under strain from 71 to 3128 με, which is higher than eight-layer strain sensor (GF = 1.93) due to the uniform surface morphology and stable resistance. The strain sensors exhibited a highly linear and reversible behavior under strain of 71 to 3128 με, so that the microstrain level could be clearly distinguished. The technology of the fully inkjet-printed CNT-based microstrain sensor provides high reproducibility, stability, and rapid hardness detection.

**Keywords:** inkjet printing; carbon nanotubes; strain sensors

## 1. Introduction

Strain sensors have great potential for application in wearable electronic devices and smart skin to monitor human motion, personal health monitoring, and human–machine interfaces. Strain sensors convert mechanical deformation into electrical signals such as capacitance or resistance to monitor changes in strain, force, pressure, and shape. Carbon nanotubes (CNTs) possess impressive properties, such as a hundred times higher tensile strength than steel, low electrical conductivity, and good thermal conductivity [1,2]. Many researchers have investigated CNTs as pressure sensors [3,4] and strain sensors [5,6] because of their excellent electromechanical properties. Bu et al. reported drop-casting-prepared multi-walled CNT films with a sensitivity of 2.5 for the strain below 0.1% [7]. Lee et al. proposed a spray-coated single-walled CNT film with good linearity over a microstrain range of 0 to 400 $\mu\varepsilon$ [8]. Sahatiya et al. deposited multi-walled CNTs on an eraser by rod-coating and obtained a strain-and-pressure sensor with a strain gauge factor (GF) of 2.4 and capacitive pressure sensor sensitivity of 0.135 MPa$^{-1}$ [9]. Chun et al. reported highly oriented and free-standing hydrophobic CNT sheets synthesized by chemical vapor deposition and transferred onto polyethylene naphthalate substrate; the CNT sheets exhibited sensitive piezoresistive responses to applied pressures of 0.1–40 kPa [10]. Previous studies have presented the fabrication technologies of CNT films such as drop-casting, spray coating, rod-coating, and chemical vapor deposition, which are complex and laborious processes results in high manufacturing costs and generate large material waste. In addition, effectively controlling the precise amount of sensing materials using the above processes is difficult.

Inkjet printing technology is a fast, convenient, low-fabrication-temperature, and low-cost technique. The ability of inkjet printing is to directly print patterns layer-by-layer based on digital images to provide additive manufacturing, without involving any photolithography process [11–14]. Moreover, inkjet-printed technology can be claimed to be environmentally friendly due to the savings in materials and the reduction in the amount of waste generated which provides more economical than other additive manufacturing technologies [15–17]. Filling-in an A4 size paper only requires 0.75 mL of silver ink, about U.S. $7.5. The filling-in an A4 paper is about 2 h with a frequency of 20 kHz. Carbon nanotubes are common materials used for printing on various substrates based to obtain accurate and controllable conductive circuits and sensors [18–21]. A counterfeit fingerprint with electrical and geometric patterns of exquisite ridges and furrows of one to five layers was printed on paper using an inkjet printer; the printed patterned CNT film electrode as the top plate of the capacitive sensor could unlock a smartphone [22]. Some researchers in a previous study fabricated an inkjet-printed 20-layer CNT-based strain sensor with a 4.7% standard deviation over 20 cycles for strains between 0 and 800 $\mu\varepsilon$ [23]. An inkjet-printed carbon nanotube embedded thin film is able to capture non-uniform strain distribution at 0.001% strain level [24]. Therefore, the inkjet printing technology can be precisely controlled the sensing material based on the ink content or number of layers. However, CNT ink is difficult to disperse CNTs due to van der Waals forces between CNTs. The efficiency of CNT dispersion in the ink affects the uniformity of CNT films and even cause nozzle clogging. Inkjet-printed CNT film is easy to agglomerate. Agglomerates CNT films deteriorate the electrical and mechanical properties because of microscopic inhomogeneity. CNTs can agglomerate at the periphery during inkjet printing (so-called the coffee ring effect), resulting in an uneven film and unstable resistance [25]. Added to the surfactant in the ink such as sodium dodecyl benzene sulfonate to improve continuity and high uniformity of the inkjet-printed CNT films was presented [26]. However, the rinsing process performed every two layers to remove most of the residual surfactant and solvent is unavoidable and time-consuming. In the current work, an aqueous conductive ink of multi-walled carbon nanotubes prepared by ball-milling dispersion method was used to provide good dispersion. Multi-pass-printing and pattern rotation were used to improve the coffee ring effect and obtain stable resistance and a uniform CNT film. The morphologies and electrical characteristics of an inkjet-printed CNT film as a strain-sensing layer were studied. Two varieties of ink, namely silver nanoparticle ink and CNT ink, were effectively arranged on a substrate according to digital images. A fully inkjet-printed CNT-based microsensor was fabricated using CNTs and silver inks. The electrical and mechanical properties of

three strain sensors printed eight-layer CNT film by unidirectional printing, rotated by 180° and 90° were compared. The CNT film printed with 90° rotation provides lower resistance and better strain sensitivity for electrical and mechanical properties. The GF of 20-layer strain sensor was 2.77 under strain from 71 to 3128 με. The microstrain sensor electromechanical properties were characterized to validate its feasibility, repeatability, and reversibility. The fully inkjet-printed strain sensor with microstrain detection capability can be used to distinguish scleroderma hardness and detect thin-film cracks. The article is organized as follows: Section 2 outlines the fabrication process and experiment procedure. Section 3.1 presents the characteristics of inkjet-printed CNTs by multi-pass printing and pattern rotation. Section 3.2 focuses on the electrical properties of strain sensors using bulge test. Finally, concluding remarks are offered.

## 2. Experiments

In this study, a 100 μm CLTE-MW (Rogers Corp., Chandler, AZ, USA) was selected as the substrate. CLTE-MW substrate are ceramic filled, woven glass reinforced Polytetrafluoroethene composites according to the manufacturer datasheets they exhibit good mechanical properties for bulge test, such as flexural strength of 113 MPa, tensile strength of 83 MPa, and flex modulus of 6463 MPa [27]. The electrodes was pattern defined using silver nanoparticle ink (DGP-40LT-15C, Advanced Nano Products Co., Ltd., Sejong, Korea) and the sensing layer was fabricated using carbon nanotube ink (CNT-22, lab311, Ltd., Seoul, Korea). Silver nanoparticle ink contained 30 to 35 wt.% silver loading, dispersed in triethylene glycol monoethyl ether with a viscosity of 10–18 cP and surface tension of 30–40 dyne/cm. The viscosity and surface tension of the CNT ink were measured as 2.5 ± 0.1 cP and 40 mN/m, respectively. The CNT and silver films were printed using a Dimatix DMP 2831 printer (FUJIFILM, Valhalla, NY, USA) equipped with a DMC-11610 cartridge. The cartridge contained 16 nozzles, 21 μm in diameter and each nozzle generated 10-pL ink drops. The substrate holder comprises of a built-in heater able to heat the substrate up to 60 °C.

Figure 1 illustrates the fabrication of the fully inkjet-printed strain sensor. Before printing, the CLTE-MW surface was treated via plasma polymerization to obtain the necessary hydrophilicity for printing. The plasma activation was performed using oxygen gas at a pressure of 9.33 Pa by an radio frequency (RF) generator operating at 13.56 MHz. Two copper films on CLTE were reserved as wire-bonding pads for measurement. A five-layer silver film was printed onto the CLTE substrate using silver nanoparticle ink on a 60 °C thermal chuck and then baked on a 100 °C hard plate for 30 min to ensure drying. Subsequently, a multilayer sensing film was inkjet-printed using CNT ink at room temperature and then baked on a hard plate at 150 °C for 1 h. Multi-pass printing and pattern rotation was employed to improve the coffee ring effect and obtain a uniform CNT film. The CNT films printed by unidirectional and rotated by 180° and 90° were compared. Then, a second silver film was printed by five passes on a 60 °C thermal chuck, with 10 min delay per pass printing to avoid silver ink flowing into the CNT film. Finally, the films were dried in a baking oven at 150 °C for 2 h.

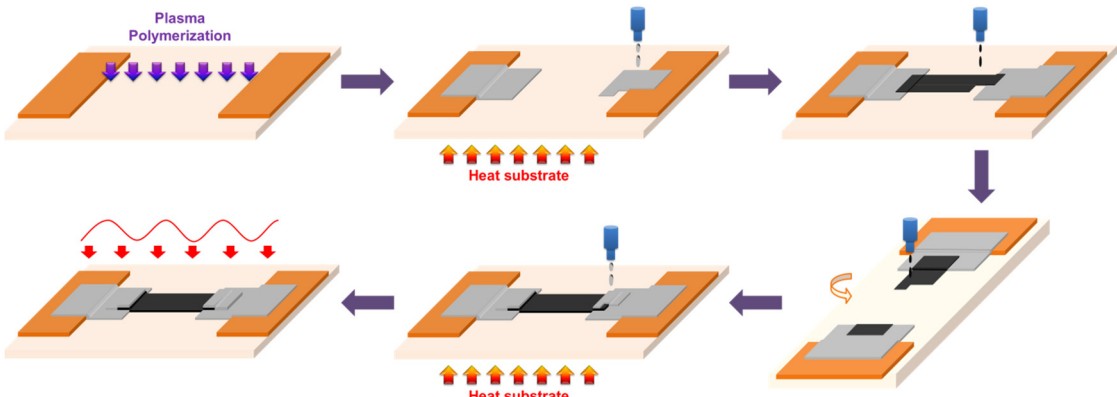

**Figure 1.** The fabrication of the fully inkjet-printed strain sensor.

Micro-Raman spectroscopy with excitation by a 532-nm green laser at a current of 0.84 A was used to study the characteristic features of inkjet-printed CNT film. Scanning electron microscopy (SEM, S3400N, Hitachi, Tokyo, Japan) under a 15-kV accelerating voltage was employed to obtain the surface morphologies and thicknesses of CNT films. Electrical resistance was measured using a digital multimeter (Rigol DM3058E, Rigol Technologies, Inc., Beaverton, NY, USA) to examine the tensile strain by the bulge test method. Three samples with a length of 3 mm and a width of 1 mm for various printing passes were measured. Sheet resistance value can be calculated by the resistance value from number of squares.

Figure 2a shows the bulge test fixture with an o-ring of 24-mm inner diameter to made the chamber airtight. Owing to the sample being face-up, the bulge test fixture was placed inside a sealed box to isolate environmental interference. Nitrogen was passed through the sealed box to maintain dryness. The pressure in bulge chamber was also controlled by introducing $N_2$ through a pressure regulating valve. The sample bulges outward when the pressure controlled by nitrogen gas is smaller than the cavity pressure. Two time controllers were used to switch pressure and maintain the pressure in a cavity for the required time. Strain ($\varepsilon$) is the relationship between the deformation and the original length of the CLTE substrate, which can be derived from the geometry relationship in Figure 2b and expressed as follows [28]:

$$\varepsilon = \frac{R\theta}{r} - 1 = \frac{2h^2}{3a^2} \tag{1}$$

where *a* is the radius of the o-ring, *h* is the height of the substrate at the center, *R* is the radius of curvature of deforming substrate, and $\theta$ is half the angle subtended by a dome surface at its center of curvature. The height was measured 10 times under various pressures using a height gauge, and the average value was taken. In this work, a maximal strain of 3128 $\mu\varepsilon$ induced a deformation of 0.822 mm.

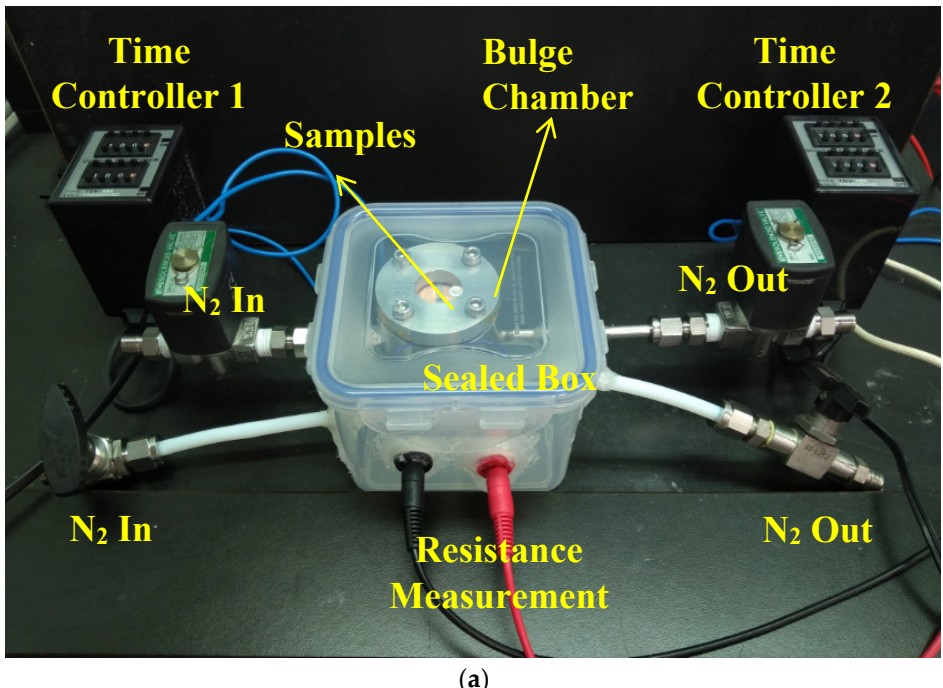

(**a**)

**Figure 2.** *Cont.*

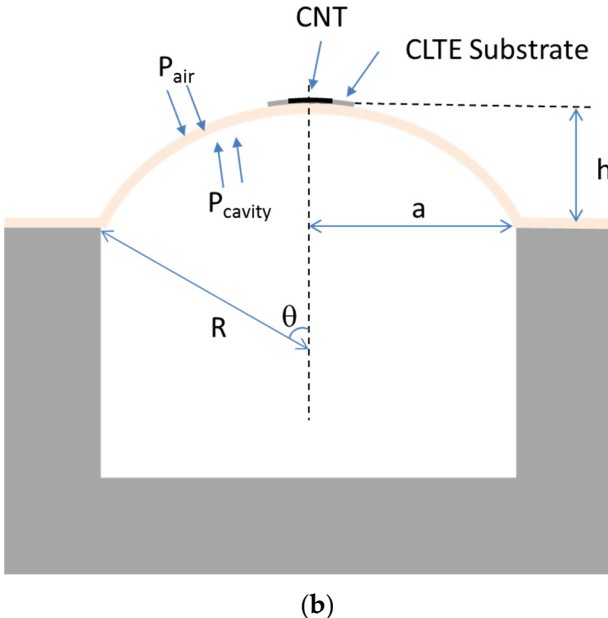

(**b**)

**Figure 2.** (**a**) Bulge test fixture; (**b**) schematic of geometry relationship for the bulge test.

## 3. Results and Discussion

### 3.1. Characteristics of Inkjet-Printed CNTs

Figure 3 shows the micro-Raman spectroscopy results of the CNT film. Three peaks of typical CNT film signatures were observed. The *D* band at 1343 cm$^{-1}$ represents the defect band, the *G* band at 1587 cm$^{-1}$ represents the graphitic band, and the peak at 2692 cm$^{-1}$ represents the *G'* band. The $I_D/I_G$ ratio was found to be 0.735, which is larger than that of the pristine CNTs because the ball-milling process used to obtain effective dispersion increases the number of defects [29].

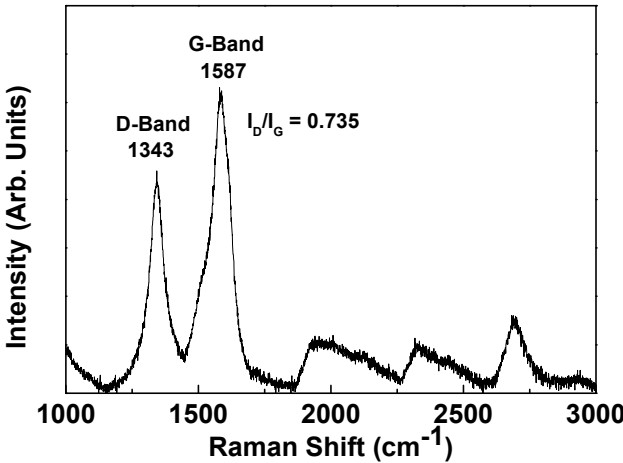

**Figure 3.** Raman measurements for inkjet-printed CNT film.

Multi-pass printing is required for optimal thickness to improve conductivity and uniformity. The common printing direction (unidirectional printing) is along the pattern from left to right and from top to bottom. Figure 4a shows an eight-layer CNT film with unidirectional printing from left to right. The CNTs were agglomerated on the left edge because of the edge-enhanced evaporation of the aqueous multi-walled CNT ink, a phenomenon referred to as the coffee ring effect. More voids were found at the right edge. The density of CNTs was decreased from the left to right side. In order

to improve coffee ring effect, multi-pass printing with pattern rotation of inkjet-printed CNT films was adopted.

Figure 4b,c show CNT films rotated by 180° and 90° after printing for every 4 and 2 passes, respectively. Comparing the surface morphologies, the CNT film printed with 180° rotation after every 2 passes featured slight agglomeration on the left and right sides. The densities of CNTs on the top and bottom sides was lower than those on the left and right sides. For 90° rotation, the densities of CNTs for all sides were similar. It indicates good uniform was obtained for the CNT film printed with 90° rotation. Figure 5 compares the sheet resistivities and thicknesses of eight-layer CNT films obtained by unidirectional printing, 180° rotation, and 90° rotation. The thickness was measured from SEM cross-section for three points from left to right side. The error bar of thickness decreases because the surface roughness decreases by pattern rotation.

The sheet resistance also decreased with a decrease in the rotation degree. This indicates that pattern rotation improved the CNT film uniformity. The thickness of eight-layer CNT film was about 1.5 μm for 90° rotation sample. The thickness and density of CNT film for unidirectional printing, 180° rotation, and 90° rotation are consistent with sheet resistivities. The pattern rotation of CNTs provides uniform films, leading to lower and stable resistance.

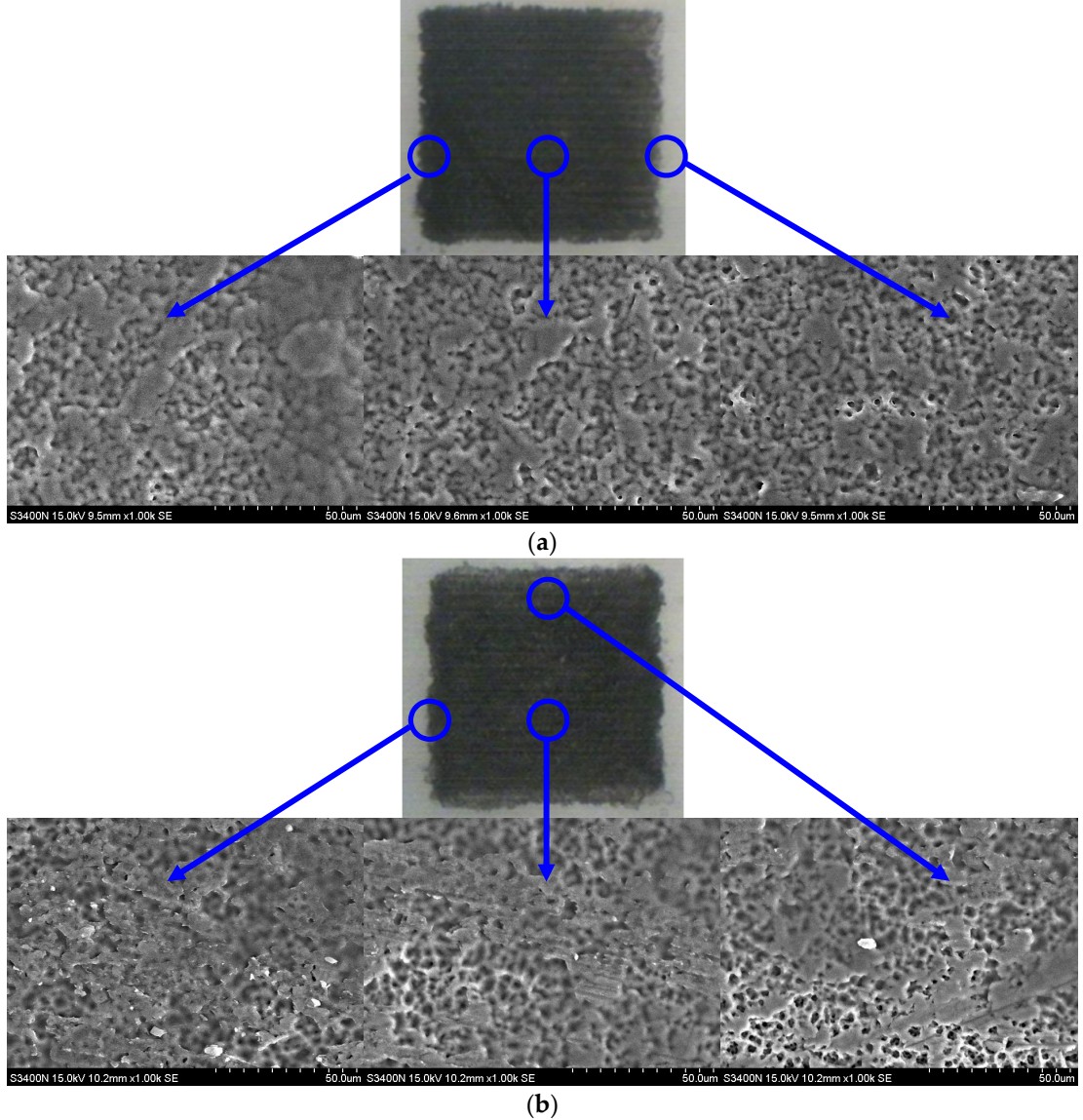

**Figure 4.** *Cont.*

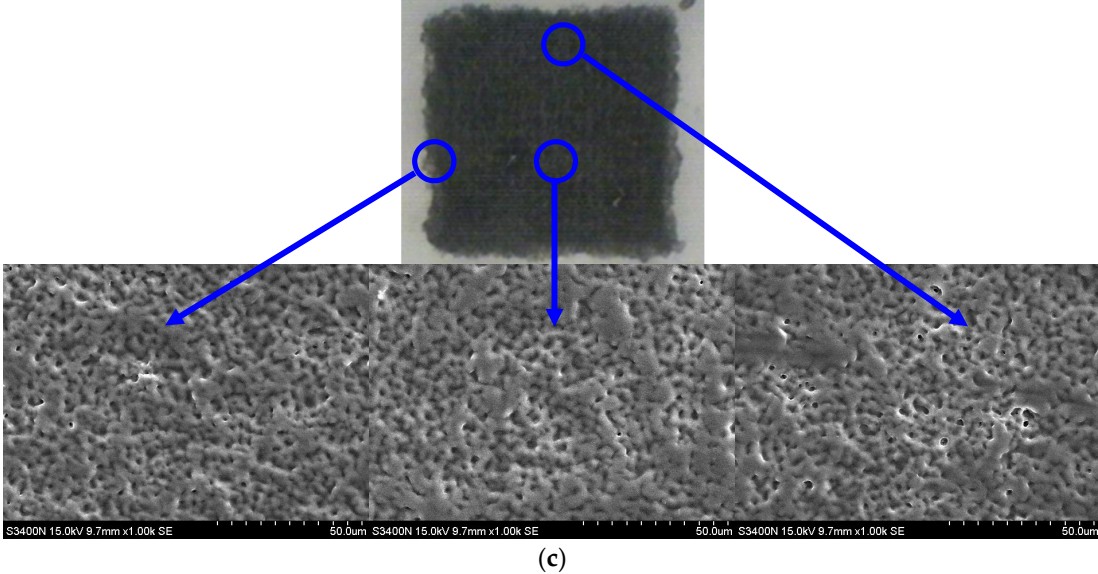

(c)

**Figure 4.** Optical photographs and surface morphologies of eight-layer CNT film with (**a**) unidirectional printing; (**b**) 180° rotation; (**c**) 90° rotation.

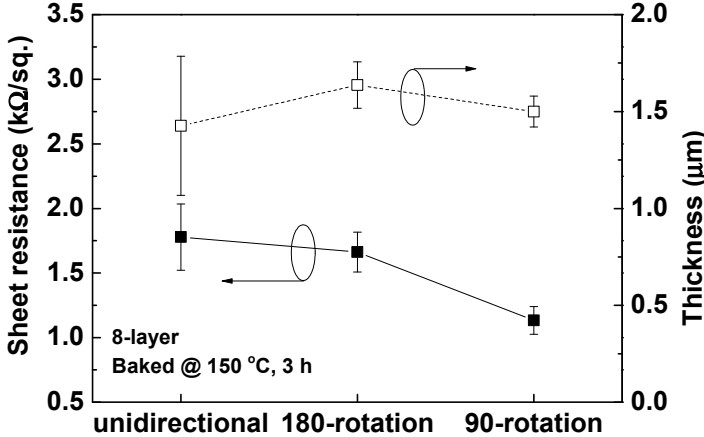

**Figure 5.** Sheet resistivities and thicknesses of eight-layer CNT films for by unidirectional printing, 180° rotation, and 90° rotation.

Figure 6 illustrates sheet resistance of 90°-rotated CNT films with respect to the number of printing passes. The number of every printing pass was calculated by dividing the total number of passes by 4. As expected, the sheet resistance decreased as the number of passes increased. A saturated sheet resistance was found after 20 layers. The sheet resistance of the CNT film was formed by thousands of individual CNTs through van der Waals forces. The increase in the number of CNT passes means an increase in the CNTs density; therefore, the sheet resistance decreased with the number of passes. The saturated sheet resistance after 20 layers indicates the more connection points between individual CNTs. Additionally, sheet resistance uniformity of the CNT films relies on the number of printing passes, as shown in the error bar. To analyze the surface uniformity, the surface morphologies of CNT films at magnifications of 7000× from 8- to 20-layer printing are displayed in Figure 7. A significant number of voids occurred on the 8-pass-printed film; the voids became less pronounced on the 20-layer film, resulting in lower sheet resistance. The 20-layer film featured a uniform surface, resulting in saturated resistance. The decreased sheet resistance is attributed to the roughness-to-thickness ratio decrease with an increase in the number of passes. The 20-layer printing was selected to provide appropriate uniformity and conductivity.

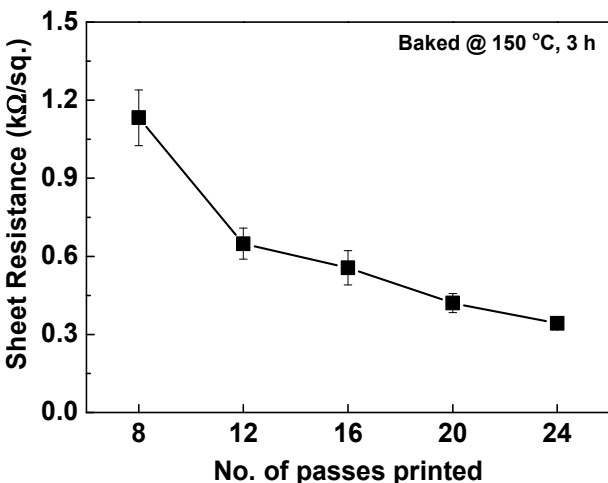

**Figure 6.** Sheet resistance of 90°-rotated CNT films with respect to the number of printing passes.

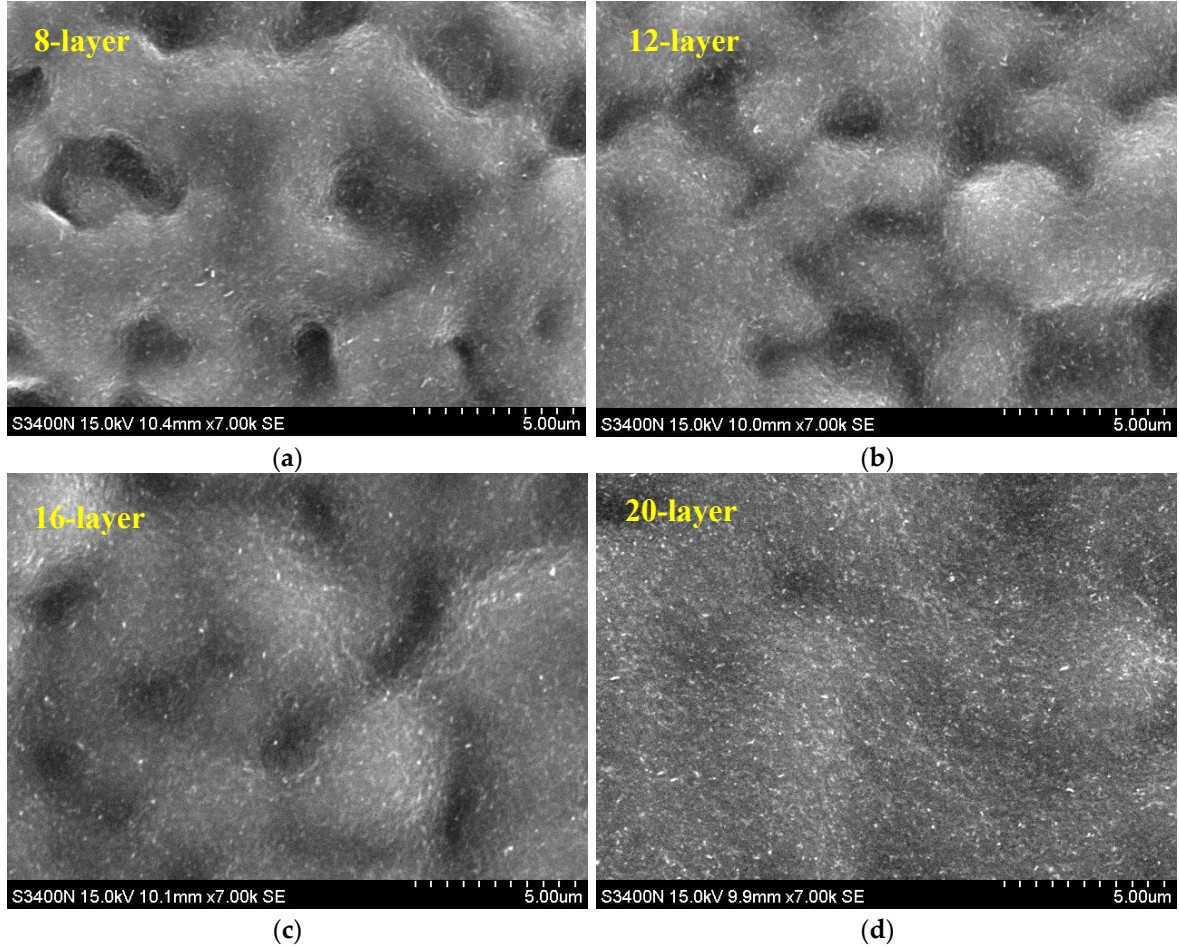

**Figure 7.** Surface morphologies of CNT films at magnifications of 7000× from (**a**) 8-layer printing; (**b**) 12-layer printing; (**c**) 16-layer printing; (**d**) 20-layer printing.

*3.2. Electrical Properties of Strain Sensors*

To investigate the sensor repeatability, multiple cycles were carried out via the bulge test using cyclic stretching and relaxation for the 20-layer fully inkjet-printed strain sensor. The sensing area had a length of 3 mm and a width of 1 mm. Figure 8 shows the transient response of the relative resistance change ($\Delta R/R_0$) for the 20-layer strain sensor under repeated stretching/releasing cycles with tensile

strain from 0 to 3128 µε, whereby each strain was held and released for 10, 30, and 60 s. The parameter $R_0$ is the initial resistance before strain. The resistance increased when tensile strain was applied because the CNTs were disconnected. The resistance could not return to its original value when the first strain was released because the CNTs network was permanently destroyed at the first strain, which resulted in resistance increase. This irrecoverable resistance is attributed to some irreversible deformation. The $\Delta R/R_0$ was almost recovered to first release for all cycles, excluding the permanent destruction, indicating that the strain sensor possessed better recoverability characteristics. In order to provide a stable baseline for strain, we first measured the 10 maximum strain cycles to be tested to exclude permanent damage. This setting was used in subsequent experiments. Then, the relative change in $\Delta R/R_0$ between the strain and subsequent release values ($|\Delta R/R_0|$) is the quantification of the number of $\Delta R/R_0$. This value is calculated to exclude continue damage of the CNTs network and substrate recovery rate after strain, as a comparison of strain strength. The average $|\Delta R/R_0|$ for 10 cycles were 0.99%, and 1.05% after a strain holding/release time of 30 and 60 s, respectively. The $|\Delta R/R_0|$ were almost similar after a strain holding/release time of 30 and 60 s, and the resistance was maintained at a constant value over a long time period (60 s), demonstrating the stability performance. However, the $|\Delta R/R_0|$ was 0.93% after a strain holding/release time of 10 s. Although the strain rate was considerably high for 10 s, the release rate was slightly low, resulting in a lower $|\Delta R/R_0|$. Therefore, we chose 30 s in subsequent experiments.

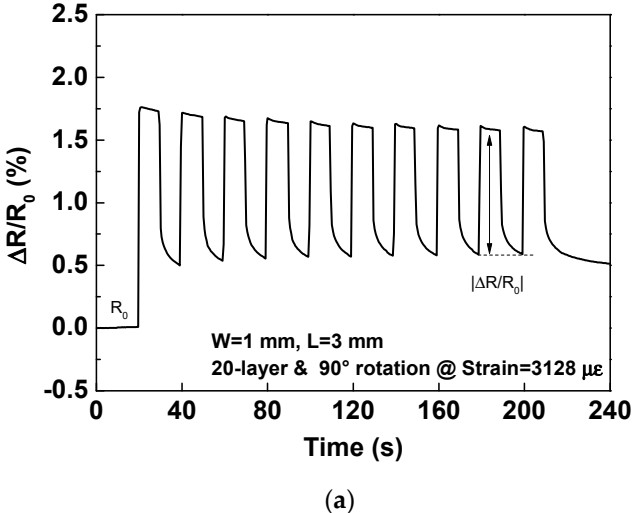

(**a**)

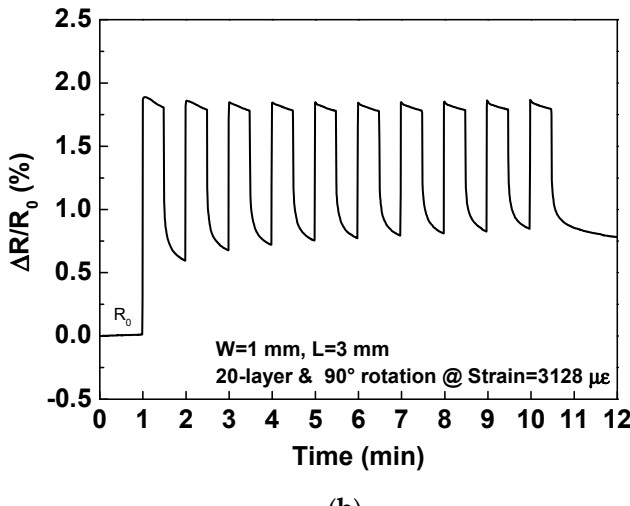

(**b**)

**Figure 8.** *Cont.*

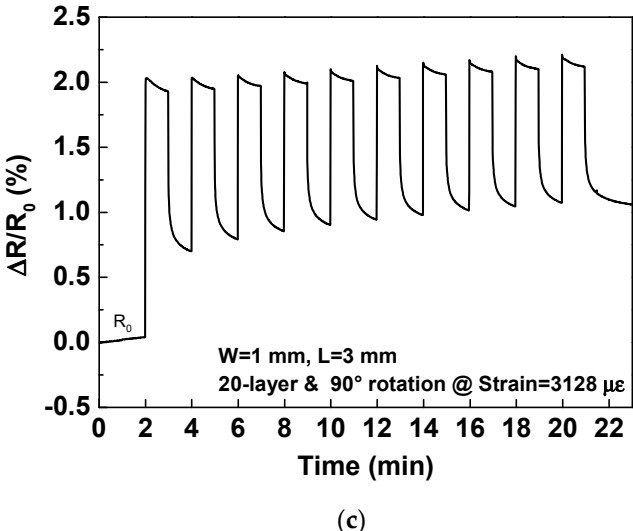

(**c**)

**Figure 8.** Transient response of the relative resistance change ($\Delta R/R_0$) for the 20-layer strain sensor under repeated stretching/releasing cycles with tensile strain from 0 to 3128 $\mu\varepsilon$, whereby each strain was held and released for (**a**) 10 s; (**b**) 30 s; (**c**) 60 s.

To examine the strain performance of the sensor excluding the permanent damage, 10 strain cycles with tensile strain from 0 to 3128 $\mu\varepsilon$ were performed before the strain sensor performance was evaluated. Figure 9a shows the stretch/release transient response as a function of various strains from 71 to 3128 $\mu\varepsilon$ for 20-layer strain sensor. $\Delta R/R_0$ was increased with tensile strain increased. $\Delta R/R_0$ after strain release was slightly increased after 3128 $\mu\varepsilon$. After the maximum strain was released, $\Delta R/R_0$ after the subsequent strain was decreased with strain decreased and could be restored to $R_0$ after 456 $\mu\varepsilon$ because the substrate recovery was slower than CNT. Figure 9b summarizes the stretch/release curves of $|\Delta R/R_0|$ and $\Delta R/R_0$ for 20-layer strain sensor. As shown, $|\Delta R/R_0|$ and $\Delta R/R_0$ increased almost linearly with an increase in the tensile strain. The release curve of $\Delta R/R_0$ is slightly larger than stretch curve due to the slow recovery of substrate. The $|\Delta R/R_0|$ values of the stretch/release curves were almost similar, indicating that the CNT film can be fully reversible by excluding substrate recovery. The GF is the ratio of fractional change in resistance versus the mechanical strain and is given by [30]:

$$\mathrm{GF} = \frac{\Delta R}{R} / \varepsilon \tag{2}$$

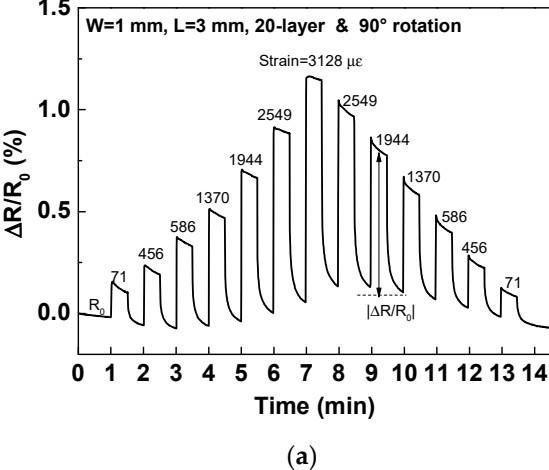

(**a**)

**Figure 9.** *Cont.*

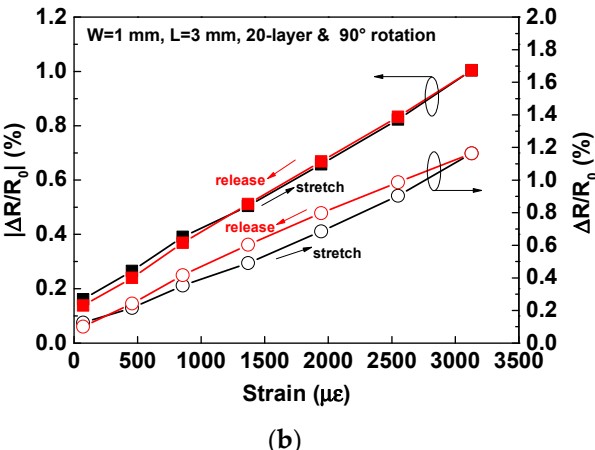

(**b**)

**Figure 9.** (**a**) Stretch/release transient response as a function of various strains from 71 to 3128 με for 20-layer strain sensor; (**b**) stretch/release curves of the 20-layer strain sensor.

From the slope of the fitted line in Figure 8b, the GF of $|\Delta R/R_0|$ and $\Delta R/R_0$ were calculated to about 2.77 and 3.42, respectively. The strain sensitivities of inkjet-printed 20-layer strain sensor is comparable to the GF of commercial strain gauges [31,32]. The strain from 71 to 3128 με could be clearly distinguished by the strain sensor, due to highly linear behavior.

Figure 10 shows the stretch/release transient response as a function of various strains from 71 to 3128 με for eight-layer strain sensor with unidirectional printing and 180° rotation. The unidirectional printing eight-layer strain sensor causes damage after tensile of 71 με. The relative resistance was larger than 3% for tensile strain from 456 to 3128 με. However, it is not the permanent destruction because the relative resistance returned to $R_0$ after each strain and the reversibility of 71 με was also found. The relative resistance increases significantly for eight-layer strain sensor with unidirectional printing when tensile strain lager than 456 με. The increase in resistance for eight-layer strain sensor is due to the CNT density on the one side by unidirectional printing is lower, resulting in more disconnection of the CNTs network and poor connectivity with silver film. For 180° rotation eight-layer strain sensors, the relative resistance could not recover to $R_0$ and the negative relative resistance was found when tensile strain lower than 456 με. This is because strain is applied to the uneven CNT film resulting in non-uniform strain distribution and permanent destruction. Some resistance singularities were measured at the moment of stretching/releasing strain because of poor connection for uneven CNT film. Non-uniform CNT film causes damage when strain is applied. Therefore, the uniformity of the CNT film cannot be ignored for the electronic and mechanical properties.

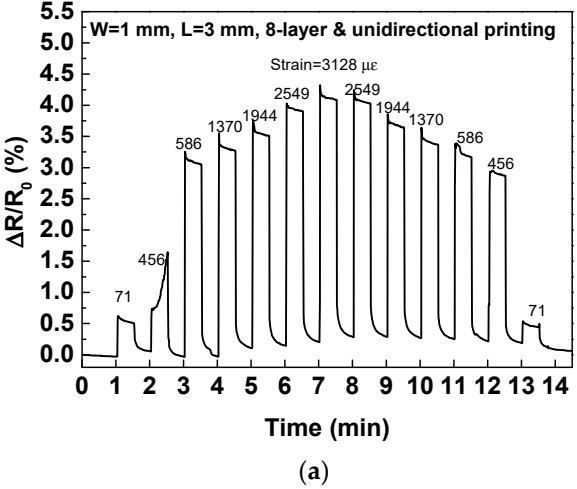

(**a**)

**Figure 10.** *Cont*.

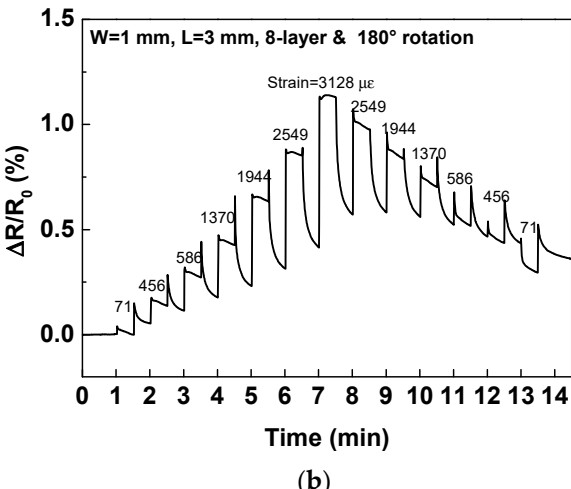

**Figure 10.** Stretch/release transient response as a function of various strains from 71 to 3128 με for eight-layer strain sensor with (**a**) unidirectional printing; (**b**) 180° rotation.

Figure 11a shows the transient response of the relative resistance change for the eight-layer with 90° rotation strain sensor under repeated stretching/releasing cycles with tensile strain from 0 to 3128 με, whereby each strain was held and released for 30 s. The resistance also increased when tensile strain was applied and could not recover to its original value, which was similar to 20-layer strain sensor. The average $|\Delta R/R_0|$ of 8-layer strain sensor for 10 cycles were 0.76% for a strain holding/release time of 30 s. The irrecoverable relative resistance change and average $|\Delta R/R_0|$ of 8-layer strain sensor is lower than that of 20-layer because of lower density. Figure 11b shows the stretch/release transient response as a function of various strains from 71 to 3128 με for eight-layer strain sensor with 90° rotation. $\Delta R/R_0$ was also increased with tensile strain increased and was decreased after release. The strain behaviors including reversibility, reproducibility, and stability for 8-layer strain sensor was similar to 20-layer strain sensor.

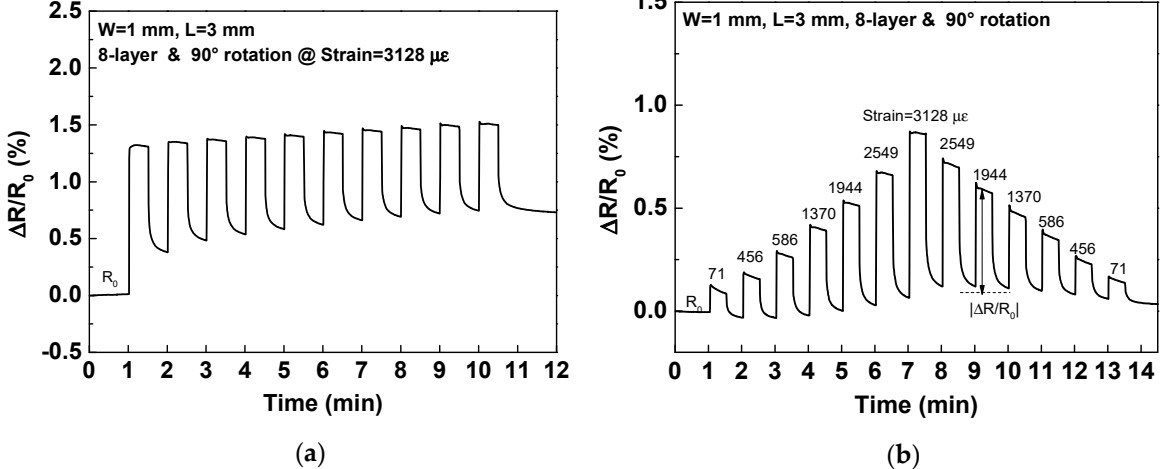

**Figure 11.** Stretch/Release transient response of the relative resistance change for eight-layer strain sensor with 90° rotation (**a**) under repeated stretching/releasing cycles with tensile strain from 0 to 3128 με, whereby each strain was held and released for 30 s; (**b**) at various strains from 71 to 3128 με.

Figure 12 compares the stretch/release curves of $|\Delta R/R_0|$ for eight-layer strain sensor by unidirectional printing, 180° rotation, and 90° rotation with tensile strain from 71 to 3128 με. The unidirectional printing strain sensor can only apply a strain of 71 με. The non-linear behavior of $|\Delta R/R_0|$ were found after 456 με. The strain sensitivities for eight-layer strain sensor with 180° and

90° rotation exhibited a linear and reversible behavior under 71 to 3128 με. However, the $|\Delta R/R_0|$ was negative while tensile strain lower than 456 με. It indicates the strain sensor with 180° rotation can only measure the tensile strain in the range of 586 to 3128 με. The GF for 180°- and 90°-rotated strain sensor were about 1.85 and 1.93, respectively. The lower GF and smaller measured range of 180° rotation strain sensor was due to uneven CNT film. The strain distribution in an unidirectional printing and 180° rotation strain sensors was non-uniform because of uneven CNT film, which decreases the linearity, sensitivity and measurement range of the sensor. It indicates that the uniformity of CNT film affects the mechanical properties. The uniformity of the CNT film was closely related to the electrical and mechanical properties. In addition, the GF of eight-layer strain sensor was lower than that of 20-layer strain sensor because of lower CNT density and higher resistance. Higher multi-pass printing provides a uniform surface resulting in better strain behaviors, which was consistent with surface morphologies and sheet resistances. The results demonstrate the mechanical deformation and electrical resistance of the fully inkjet-printed CNT-based strain sensor; the sensor features high reversibility, reproducibility, and stability to afford microstrain detection and classification. Fully inkjet-printed CNT film with multi-pass printing and pattern rotation provides better and stable mechanical performance for strain applications.

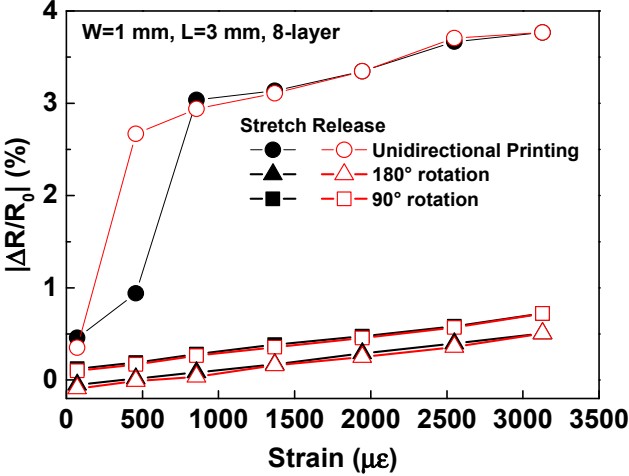

**Figure 12.** Stretch/release curves of the eight-layer strain sensors by unidirectional printing, 180° rotation, and 90° rotation.

## 4. Conclusions

A fully inkjet-printed strain sensor was successfully fabricated using a CNT-sensing film and a silver conductive film. Based on the digital image of the inkjet printing technology, a uniform and stable-resistance CNT film was obtained via multi-pass printing and pattern rotation. Inkjet-printed CNT and silver films were fabricated layer-by-layer to form strain sensors. The uneven carbon nanotube film causes poor electrical and mechanical properties of strain sensors. The low strain behaviors such as non-linear, small measurement range, and low sensitivity were observed for strain sensor with unidirectional printing and 180° rotation because of uneven CNT film. The relative resistance change for the 8- and 20-layer strain sensor with 90° rotation was 0.76% and 1.05% for 10 cycles between 0 and 3128 με strain, respectively. The GF of 8- and 20-layer strain sensors with 90° rotation were 1.93 and 2.77, respectively. The higher GF of 20-layer strain sensor was due to higher density and uniform surface of multi-pass printing. The strain sensors with 90° rotation exhibited a linear and reversible behavior under 71 to 3128 με. The fully inkjet-printed CNT-based strain sensor showed high reversibility, repeatability, and linearity; therefore, it can be applied to classify microstrain for distinguishing scleroderma hardness and thin-film crack detection.

**Author Contributions:** Conceptualization, H.-L.K.; methodology, H.-L.K. and C.-L.C.; validation, C.-L.C. and Y.-C.T.; resources, H.-L.K. and L.-C.C.; writing—original draft preparation, H.-L.K. and C.-L.C.; writing— review and editing, L.-C.C., C.-B.C., and W.-H.C.; funding acquisition, H.-L.K., C.-B.C., and W.-H.C. All authors have read and agreed to the published version of the manuscript.

**Funding:** This work was supported by the Chang Gung Memorial Hospital, Taiwan (CMRPD2H0311 & BMRP957).

**Acknowledgments:** The authors thank the Centre for Reliability Sciences and Technologies at Chang Gung University for their help.

**Conflicts of Interest:** The authors declare no conflict of interest.

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
