# Peer review of "A Fully Inkjet-Printed Strain Sensor Based on Carbon Nanotubes"

_coatings, doi:10.3390/coatings10080792_

Round 1

Reviewer 1 Report

In this paper, the authors presented carbon nanotube-based strain sensors that were fabricated by inkjet-printing. The authors claimed that uniform CNT films and stable film resistance can be achieved by multi-pass printing and pattern rotation. The manuscript may need major revision to be publishable. An Inkjet-printed CNT-based film has been extensively studied in other papers (e.g., Highly reproducible, hysteresis-free, flexible strain sensors by inkjet printing of carbon nanotubes, Carbon, 95, 1020-1026 (2015); Inkjet-printed stretchable single-walled carbon nanotube electrodes with excellent mechanical properties, Applied Physics Letters, 104, 113103, (2014); An inkjet-printed carbon nanotube strain distribution sensor for quasi real-time strain monitoring of lightweight design materials, Advances in Science and Technology, 101, 3-8, (2016); ). Therefore, the author should be able to provide substantial new knowledge and information that significantly advance the field. In addition, 1. Introduction section should be improved to give sufficient background to the readers. My comments and suggestions to the authors are shown below:

  1. The authors are needed to provide new knowledge and information to the readers.
  2. [Page 1-2] Introduction should be improved providing sufficient background of the area with enough references. In addition, it is recommended to have one main idea per paragraph. Several main ideas were made in the introduction, which can confuse the readers.
  3. [Page 4, line 108] The authors claimed that “multi-pass printing is required for optimal thickness to improve conductivity and uniformity”. However, thickness of films were not presented in the manuscript, and it is necessary to provide thickness results of the eight-layer CNT films (unidirectional, 180-rotation, and 90-rotation) at multiple points of the specimen to support the claim.
  4. Multiple samples along the direction of the printed direction (
  5. [Page 4, line 114 & Figure 4 (a-c)] Figure 4 (a-c) is not sufficient to support the authors claim in line 114-116. The authors are suggested to provide micro-scale images to show CNT agglomerations and support the claim.
  6. 2. Experiments section should be improved by describing all the experimental methods that were used to obtain the results in the 3. Results and Discussion section. For example, the authors should provide information on how the sheet resistance was measured/calculated.
  7. [Page 5, line 136, Figure 4(b)] The authors should show error bars in the figure.
  8. [Page 8, line 178] Only one specimen was tested to evaluate strain sensing performance of the CNT film. It is necessary to evaluate and compare CNT films fabricated with all inkjet-printed conditions (unidirectional, 180-rotation, and 90-rotation).

Author Response

The valuable comments and suggestions from the Editors and reviewers have been incorporated in the revised manuscript. Attached file is point-by-point responses to the reviewers' comments. We would like to thank the editors and reviewers sincerely for the time and effort spent to help improve the presentation of this paper. Please see the attachment.

Reviewer 2 Report

  1. Good idea, good work done
  2. Please explain what is CLTE-MW substrate (page 3 line 68)
  3. Please correct ..."plasma polymerization... page 3 line 72. Probably the Authors mean ..oxygen plasma activation... because there is no monomer added for polymerization
  4. Figure 2 - where was the tester mounted ? On the top of bulge curvature ? Please mark it.
  5. Please add error bars in Figure 3 (ther are correctly added in Figure 4)
  6. In Fig.2 there is depicted "air pressure". Was it really an air ? There are two nitrogen inlets, two nitrogen outlets and it suggests, that all the experiments were done in N2 atmosphere.
  7. All points above are of small importance, easy to correct or even skip. For me important point is to compare perforance of used MWCNT sensor with other commercial product. The name of commpercial product can be hidden or skipped but it will show real importance of done work. Please consider if the Authors can add such a comparison.

Author Response

The valuable comments and suggestions from the Editors and reviewers have been incorporated in the revised manuscript. Attached file is point-by-point responses to the reviewers' comments.We would like to thank the editors and reviewers sincerely for the time and effort spent to help improve the presentation of this paper. Please see the attachment. 

Reviewer 3 Report

Paper is well-written. Results are clearly presented. 

-A more detailed introduction on CNT printing technology could be useful. 

-An explicit analysis of cost with respect to commercial alternatives may be great.

-In Fig. 7, baseline for relative resistance is varying. Is there any solution to prevent it, maybe is it because of heating because of repeated stretching/releasing or structure of sensor is distorted?

Good work.

Round 2

Reviewer 1 Report

The manuscript can be published in the pesent form.